# Associations of maternal quitting, reducing, and continuing smoking during pregnancy with longitudinal fetal growth: Findings from Mendelian randomization and parental negative control studies

Judith S. Brand[1,2,3], Romy Gaillard[4,5], Jane West[2,3,6], Rosemary R. C. McEachan[6], John Wright[6], Ellis Voerman[4,5], Janine F. Felix[4,5,7], Kate Tilling[2,3], Deborah A. Lawlor[2,3,8]*

1 Clinical Epidemiology and Biostatistics, School of Medical Sciences, Örebro University, Örebro, Sweden, 2 MRC Integrative Epidemiology Unit, University of Bristol, Bristol, United Kingdom, 3 Population Health Science, Bristol Medical School, University of Bristol, Bristol, United Kingdom, 4 Generation R Study Group, Erasmus University Medical Center Rotterdam, Rotterdam, the Netherlands, 5 Department of Pediatrics, Sophia Children's Hospital, Erasmus University Medical Center Rotterdam, Rotterdam, the Netherlands, 6 Bradford Institute for Health Research, Bradford Royal Infirmary, Bradford, United Kingdom, 7 Department of Epidemiology, Erasmus University Medical Center Rotterdam, Rotterdam, the Netherlands, 8 National Institute for Health Research Bristol Biomedical Research Centre, Bristol, United Kingdom

* d.a.lawlor@bristol.ac.uk

**Data Availability Statement:** The data underlying the results presented here are from 2 different

## Abstract

### Background

Maternal smoking during pregnancy is an established risk factor for low infant birth weight, but evidence on critical exposure windows and timing of fetal growth restriction is limited. Here we investigate the associations of maternal quitting, reducing, and continuing smoking during pregnancy with longitudinal fetal growth by triangulating evidence from 3 analytical approaches to strengthen causal inference.

### Methods and findings

We analysed data from 8,621 European liveborn singletons in 2 population-based pregnancy cohorts (the Generation R Study, the Netherlands 2002–2006 [$n$ = 4,682]) and the Born in Bradford study, United Kingdom 2007–2010 [$n$ = 3,939]) with fetal ultrasound and birth anthropometric measures, parental smoking during pregnancy, and maternal genetic data. Associations with trajectories of estimated fetal weight (EFW) and individual fetal parameters (head circumference, femur length [FL], and abdominal circumference [AC]) from 12–16 to 40 weeks' gestation were analysed using multilevel fractional polynomial models. We compared results from (1) confounder-adjusted multivariable analyses, (2) a Mendelian randomization (MR) analysis using maternal rs1051730 genotype as an instrument for smoking quantity and ease of quitting, and (3) a negative control analysis comparing maternal and mother's partner's smoking associations. In multivariable analyses,

studies with different data publishing policies. The individual-level data are not available through a public repository due to ethical and legal restrictions. Both studies have an open collaboration policy and the datasets generated and analysed for this study are available upon request. Applications to use the BiB dataset can be made directly to the BiB executive group (https://borninbradford.nhs.uk/research/how-to-access-data/) and applications to use the GenR dataset can be made to the Generation R Study Management Team (generationr@erasmusmc.nl), subject to local rules and regulations.

**Funding:** The Born in Bradford study (BiB) receives core funding from the Wellcome Trust (WT101597MA) a joint grant from the UK Medical Research Council (MRC) and UK Economic and Social Science Research Council (ESRC) (MR/N024397/1) and the National Institute for Health Research (NIHR) under its Collaboration for Applied Health Research and Care (CLAHRC) for Yorkshire and Humber and the NIHR Clinical Research Network (CRN). The general design of the Generation R Study (GenR) is made possible by financial support from the Erasmus University Medical Center, Rotterdam, Erasmus University, Rotterdam, the Netherlands Organisation for Health Research and Development (ZonMw), the Netherlands Organisation for Scientific Research (NWO), the Dutch Ministry of Health, Welfare and Sport and the Dutch Ministry of Youth and Families. This project also received funding from the European Union's Horizon 2020 research and innovation programme under grant agreements No 633595 (DynaHEALTH) and No 733206 (LIFECYCLE). The research presented in this paper is supported by the British Heart Foundation (CS/16/4/32482 and AA/18/7/34219), US National Institute of Health (R01 DK10324), the European Research Council under the European Union's Seventh Framework Programme (FP7/2007-2013) / ERC grant agreement No 669545 (the latter also provides J.S.B.'s salary), the European Union's Horizon 2020 research and innovation programme under grant agreement No 733206 (LIFECYCLE) and the NIHR Biomedical Centre at the University Hospitals Bristol, the NHS Foundation Trust and the University of Bristol. JSB., KT. and DAL work in a unit that receives UK Medical Research Council (MRC) funding (MC_UU_00011/3 and MC_UU_00011/6) and DAL is an NIHR senior investigator (NF-SI-0611-10196). RG received funding from the Dutch Heart Foundation (2017T013), the Dutch Diabetes Foundation (2017.81.002) and the Netherlands Organisation for Health Research and Development (ZonMw) (543003109). The funders had no role in the

women who continued smoking during pregnancy had a smaller fetal size than non-smokers from early gestation (16–20 weeks) through to birth ($p$-value for each parameter < 0.001). Fetal size reductions in continuing smokers followed a dose-dependent pattern (compared to non-smokers, difference in mean EFW [95% CI] at 40 weeks' gestation was −144 g [−182 to −106], −215 g [−248 to −182], and −290 g [−334 to −247] for light, moderate, and heavy smoking, respectively). Overall, fetal size reductions were most pronounced for FL. The fetal growth trajectory in women who quit smoking in early pregnancy was similar to that of non-smokers, except for a shorter FL and greater AC around 36–40 weeks' gestation. In MR analyses, each genetically determined 1-cigarette-per-day increase was associated with a smaller EFW from 20 weeks' gestation to birth in smokers ($p$ = 0.01, difference in mean EFW at 40 weeks = −45 g [95% CI −81 to −10]) and a greater EFW from 32 weeks' gestation onwards in non-smokers ($p$ = 0.03, difference in mean EFW at 40 weeks = 26 g [95% CI 5 to 47]). There was no evidence that partner smoking was associated with fetal growth. Study limitations include measurement error due to maternal self-report of smoking and the modest sample size for MR analyses resulting in unconfounded estimates being less precise. The apparent positive association of the genetic instrument with fetal growth in non-smokers suggests that genetic pleiotropy may have masked a stronger association in smokers.

## Conclusions

A consistent linear dose-dependent association of maternal smoking with fetal growth was observed from the early second trimester onwards, while no major growth deficit was found in women who quit smoking early in pregnancy except for a shorter FL during late gestation. These findings reinforce the importance of smoking cessation advice in preconception and antenatal care and show that smoking reduction can lower the risk of impaired fetal growth in women who struggle to quit.

## Author summary

### Why was this study done?

- Maternal smoking during pregnancy is an established risk factor for low infant birth weight. Understanding when and which parameters of fetal growth are affected by different smoking behaviours is important for strengthening and focusing clinical and public health guidelines.

- The importance of smoking cessation in early pregnancy and the extent to which fetal growth restriction can be prevented or minimised by lowering cigarette consumption in women who find quitting difficult is also uncertain.

### What did the researchers do and find?

- We analysed data from 8,621 white European liveborn singletons from 2 population-based pregnancy cohorts to assess the associations of maternal quitting, reducing, and continuing smoking during pregnancy with the longitudinal growth of different fetal

design of the study, the collection, analysis, or interpretation of the data; the writing of the manuscript, or the decision to submit the manuscript for publication.

**Competing interests:** I have read the journal's policy and the authors of this manuscript have the following competing interests: DAL has received support from several national and international government and charitable funders, and from Medtronic Ltd and Roche Diagnostics for research unrelated to that presented here. The other authors report no conflicts.

**Abbreviations:** AC, abdominal circumference; BiB, Born in Bradford study; CRL, crown–rump length; EFW, estimated fetal weight; FL, femur length; GenR, Generation R Study; HC, head circumference; MR, Mendelian randomization; OR, odds ratio.

parameters (weight, head circumference, femur length, and abdominal circumference). We compared results across 3 different analytical approaches (conventional multivariable, Mendelian randomization, and parental negative control analyses) to strengthen confidence in our findings.

- We found that pre-pregnancy smokers who continued smoking during pregnancy had a reduced fetal size from early gestation (12–16 weeks) onwards. Associations of maternal smoking with each fetal parameter followed a dose-dependent pattern, with fetal size reductions increasing in magnitude with the number of cigarettes smoked.

- While all fetal parameters were affected in women who continued smoking during pregnancy, size reductions were most pronounced for femur length. In pre-pregnancy smokers who gave up smoking early in pregnancy, no overall growth deficit was observed, except for a smaller femur length towards the end of pregnancy.

- The association of maternal smoking with reduced fetal growth was consistent across all 3 methods, thus providing stronger support that the association is causal, in comparison to current evidence, which relies solely on multivariable regression.

## What do these findings mean?

- Our findings reinforce existing advice promoting and supporting smoking cessation in preconception and antenatal care services; they provide strong support for these recommendations.

- The consistent results across methods for a linear dose-dependent association of maternal smoking with reduced fetal growth from early gestation in women who continue smoking during pregnancy provide evidence to support reducing smoking amounts in those who struggle to quit.

## Introduction

Maternal smoking during pregnancy is one of the most important modifiable determinants of low infant birth weight and other adverse perinatal outcomes [1]. Whilst the prevalence of pregnancy smoking has declined, it remains high in the US and Western Europe, where approximately 15%–20% of all pregnant women smoke [2]. According to the US Surgeon General's reports [3,4], there is substantial evidence supporting a direct link between pregnancy smoking and low infant birth weight, with this evidence being consistent across a multitude of studies using conventional multivariable analyses of observational data. However, these results from conventional observational approaches may be explained by residual confounding given that women who smoke during pregnancy are more likely to be socioeconomically disadvantaged and to engage in other risky health behaviours that may lead to low birth weight. More recent findings from quasi- experimental studies comparing populations with different tobacco control policies [5–7], and from observational studies using Mendelian randomization (MR) [8], parental negative control [9], and discordant sibling [10] designs, support an intra-uterine effect of maternal smoking during pregnancy on infant birth weight.

Still, evidence on the critical smoking exposure window and timing of smoking-related fetal growth restriction is limited. Previous studies have mainly focused on size at birth only, with few studies exploring when during pregnancy smoking starts to affect growth, or whether there are differences in its effect on different fetal growth parameters. Pregnancy smoking has been associated with a reduction in first trimester crown–rump length (CRL) [11], but studies using repeat ultrasound measures have failed to identify an association with early second trimester fetal size, and the magnitude of reported associations with different fetal parameters later in pregnancy varies considerably [12]. This inconsistency largely reflects between-study heterogeneity in longitudinal fetal growth assessment and adjustment for confounding factors, which hamper the ability to determine causal intrauterine effects from early fetal life [12]. There is also uncertainty about the impact of smoking cessation in early pregnancy and the extent to which fetal growth restriction can be prevented or minimised by lowering cigarette consumption. Obtaining stronger causal evidence of when in pregnancy smoking influences fetal growth, how quitting affects this, and whether there is a dose response in those who continue smoking will provide stronger evidence to inform updated guidelines and to help women make more informed decisions. Furthermore, understanding whether smoking reduces birth weight by universally reducing growth across all growth parameters or has varying effects could provide insights into the mechanisms through which smoking affects fetal growth.

The aim of this study was to determine the associations of maternal quitting, reducing, and continuing smoking during pregnancy with longitudinal trajectories of different fetal growth parameters (head circumference [HC], femur length [FL], abdominal circumference [AC], and estimated fetal weight [EFW]) in a joint analysis of data from 2 population-based pregnancy cohorts. To improve causal inference, we triangulated findings from 3 approaches with differing sources of bias (multivariable regression, MR, and parental negative control) [13]. This study provides novel insights into the impact of quitting or reducing smoking during pregnancy on fetal growth that can be used to tailor advice and support to individual women.

## Methods

### Cohorts

We identified cohorts from the MR-PREG consortia, a collaboration of cohorts used to explore causes and consequences of different pregnancy complications and outcomes. To contribute to this study, cohorts had to have repeat fetal ultrasound scan measurements. We used parental and offspring data from the Generation R Study (GenR) [14] and the Born in Bradford study (BiB) [15], 2 population-based pregnancy cohorts including participants from multi-ethnic urban populations (see S1 Text for a detailed description of both cohorts). GenR is based in Rotterdam (the Netherlands) and consists of 9,778 women (response rate at baseline = 61%) who had an expected delivery date between April 2002 and January 2006. Most participants (n = 8,880, 91%) were recruited during pregnancy. BiB enrolled 12,450 women residing in Bradford (response rate > 80%), a city in the north of England. Women participating in BiB had an expected delivery date between March 2007 and December 2010 and were mainly recruited at their oral glucose tolerance test (OGTT) appointment at 26–28 weeks' gestation (all pregnant women booked to deliver in Bradford are offered an OGTT). For the present analysis, we included women who gave birth to singletons without known fetal/birth congenital anomalies and with fetal growth and maternal smoking data (see S1 Fig). We further restricted analyses to participants of white European origin, as smoking behaviours and fetal growth trajectories differ considerably by ethnicity, and for other ethnic groups, numbers were too small for reliable estimates. This resulted in a study population of 8,621 liveborn singletons

(GenR, *n* = 4,682; BiB, *n* = 3,939). Of these, 6,527 had maternal genotype data (GenR, *n* = 3,604; BiB, *n* = 2,923) and were included in the MR analyses, and 5,537 had partner smoking data (GenR, *n* = 4,206; BiB, *n* = 1,331) for inclusion in the negative control study.

All study participants gave written informed consent, and the study was approved by the local medical ethical committees (GenR: MEC 198.782/2001/31; BiB: ref 06/Q1202/48).

### Parental smoking

Full details of how parental smoking was assessed are provided in S1 Text. Based on the available data in both cohorts, maternal smoking during pregnancy was categorised as follows: (1) non-smokers (i.e., women who had never smoked or stopped smoking several months [>6 months in GenR and >3 months in BiB] prior to pregnancy); (2) pre-pregnancy smokers who quit in early pregnancy (i.e., women who reported smoking in the months before becoming pregnant but not after the first trimester); and (3) pre-pregnancy smokers who continued smoking through pregnancy (i.e., women who reported smoking in the months before becoming pregnant, in the first trimester, and during mid and/or late pregnancy).

We also extracted information on the number of cigarettes smoked by women who continued to smoke through pregnancy. This was done by taking the mean of the smoking quantity reported during the first trimester and during mid and/or late pregnancy. Since each cohort used different cutoffs to categorise self-reported smoking quantity (intensity), these data were grouped into 'light' (1–4 [GenR] or 1–5 [BiB] cigarettes per day), 'moderate' (5–9 [GenR] or 6–10 [BiB] cigarettes per day), and 'heavy' smoking (≥10 [GenR] or >10 [BiB] cigarettes per day).

To strengthen causal inference, we used information on partner smoking during pregnancy as a negative control [13,16]. Under the assumption that confounding would be similar for maternal and partner smoking, stronger estimates for the association of fetal growth parameters with maternal than partner smoking can be interpreted as support for a causal intrauterine effect of maternal smoking, while similar effect estimates would be suggestive of unmeasured shared familial confounding [13]. We used the same categories of partner smoking quantity (light, moderate, and heavy smoking) to those used for maternal smoking.

### Instrument selection and genotyping for MR

We selected the rs1051730 single nuclear polymorphism (SNP) in the α-nicotinic acetylcholine receptor (CHRNA3/5) gene cluster as an instrumental variable for our MR analysis because this variant has been shown to robustly (in genome-wide discovery and replication studies [17,18]) relate to smoking intensity and ease of quitting, and also to relate to these smoking traits in pregnancy [19]. Full details of genotyping and quality control are provided in S1 Text and S1 Table.

### Fetal growth assessment

Longitudinal fetal growth trajectory analyses were based on repeat fetal ultrasound and birth anthropometric measurements starting at 12–16 weeks' gestation and ending at term. The collection of fetal ultrasound data in both cohorts has been described in detail elsewhere [20,21]; see also S1 Text for study-specific procedures including numbers of anthropometric measurements collected. In GenR and BiB, gestational age was determined using CRL (up to 13 weeks 6 days) and biparietal diameter thereafter. Fetal anthropometrics (HC, FL, and AC) were collected from 12 weeks (HC and FL) and 16 or 18 weeks (AC) of gestation and measured to the nearest millimetre using standard ultrasound planes [22]. From these measurements, fetal

weight was estimated using the Hadlock 1985 formula [23]:

$$\log_{10} \text{EFW} = 1.326 - 0.00326 \text{AC} \times \text{FL} + 0.0107 \text{HC} + 0.0438 \text{AC} + 0.158 \text{FL}$$

Anthropometric measurements at birth (HC and weight in both cohorts and AC in BiB only) were obtained from obstetric medical records and combined with fetal anthropometric data to estimate trajectories of fetal growth. In both cohorts, the median number of repeat weight estimates per singleton was 3.

### Other variables

The following variables were considered as potential confounders in multivariable analyses, including for partner smoking associations: age, parity (at the time of index pregnancy; mothers only), height, body mass index, education, and alcohol intake. In both cohorts, these variables were mainly derived from the baseline questionnaire administered at enrolment at a median gestational age of 13 weeks (GenR) or 26 weeks (BiB). Full details of their assessment and inclusion in the analysis models are provided in S1 Text.

### Statistical analysis

A draft analysis plan was written by JSB, KT, and DAL in December 2017, and the final analysis plan was agreed upon by all investigators in February 2018 after discussions with the GenR team (see S2 Text). Analyses commenced in June 2018, and 2 changes were made after analyses had begun. Preliminary results showed that maternal rs1051730 genotype was associated with maternal age, and we decided to explore the impact of this by repeating all MR analyses with adjustment for maternal age. In January 2019, in response to suggestions from one of the co-authors, it was agreed that we would also estimate and report proportional differences in mean fetal size as we might expect absolute differences to increase as the fetus grows. No further changes to the analysis plan were made.

Fetal growth trajectories were derived using multilevel fractional polynomial models with 2 levels (i.e., measurements within occasions [level 1] within individuals [level 2]) as described in detail elsewhere [21]. A variable for cohort (GenR versus BiB) and one for its interaction with gestational age were entered in all models to account for between-study differences in fetal growth. In pooling the data in a single analysis model, we assume that both cohorts are from the same underlying population for which inferences can be made. To test this assumption, we compared estimates obtained from this single analysis model with those observed in GenR and BiB separately and further tested for between-study heterogeneity by adding a 3-way interaction term (between study, smoking exposure, and gestational age) to each analysis model. More details on model specification can be found in S1 Text, including supportive data for the growth trajectories fitted (S2 and S3 Tables; S2 Fig). We also compared the fetal growth results with associations observed for birth weight using linear regression. Because fetal growth trajectories, by definition, are conditioned on gestational age, we checked the extent to which birth weight differences changed with adjustment for gestational age.

### Multivariable and parental negative control analyses

Associations of maternal and partner smoking exposures were analysed by adding these variables as main effects and as interactions with gestational age to the multilevel models. From these models, global $p$-values for these coefficients were derived to assess overall differences in fetal growth (i.e., to test the null hypothesis that the growth trajectories across pregnancy for different parameters do not differ, for example comparing smokers to non-smokers). This

global approach is recommended to avoid multiple testing. Differences in mean fetal size associated with each smoking exposure were estimated at 4-week intervals from 12–16 weeks onwards in absolute original units (i.e., millimetres and grams) and proportionally as the ratio of the observed differences to the mean at each time point. To provide an estimate of the timing of fetal growth restriction, we also report the earliest gestational age (based on 4-week intervals) at which the 95% CI for a difference in mean fetal size did not include 0. It should, however, be noted that this estimate is conservative as it is guided by statistical significance only, and the actual process of growth restriction will have started prior to this point in time.

Fetal size differences by parental (maternal/partner) smoking status were estimated using maternal/partner non-smokers as the reference group. Associations with parental smoking are presented with adjustment for cohort only (model 1) and with adjustment for cohort, infant sex, parity (for maternal smoking only), and respective parental age, height, BMI, education level, and alcohol use during pregnancy (model 2). In the parental negative control comparison, associations were additionally mutually adjusted for the smoking behaviour of the other parent (model 3).

Missing covariate data were imputed using multiple imputation stratified by cohort (see details in S1 Text), and we also present results based on complete case data.

### MR analysis

Fetal size differences by maternal rs1051730 genotype were modelled per risk allele (T) increase (i.e., assuming an additive genetic effect). As rs1051730 genotype has been associated with smoking quantity in individuals who smoke (with each T allele increase corresponding to approximately 1 additional cigarette per day [17,24]), we would anticipate an association in women continuing to smoke during pregnancy, and would expect this association to be weaker in those who quit smoking and null in non-smokers. An association with fetal growth in non-smokers would be indicative of a horizontal pleiotropic effect of the variant (i.e., not acting through smoking intensity or inability to quit smoking) and would suggest that our results in smokers may be biased [25]. In our main analyses we compared the associations of the genetic variant with fetal growth in these 3 groups (pre-pregnancy smokers who continued smoking through pregnancy, pre-pregnancy smokers who quit in early pregnancy, and non-smokers). Stratifying pre-pregnancy smokers into those quitting and continuing smoking could, however, introduce collider bias [26], as the rs1051730 T allele has previously been associated with the ability to quit smoking [19]. We therefore repeated the MR analysis in just 2 groups: pre-pregnancy smokers and non-smokers. We examined associations of maternal rs1051730 genotype with potential confounders of the smoking–fetal growth association to test the independence assumption. To address the possibility of an association of maternal rs1051730 being mediated through inheritance rather than a causal intrauterine effect, we performed a sensitivity analysis with adjustment for fetal rs1051730 genotype. Fetal genotype data were available for 4,457 singletons.

All analyses were undertaken using MLwiN version 2.4 run in Stata/MP version 15.

## Results

### Participant characteristics

Distributions of characteristics of participants included in the MR and negative control analyses were very similar to those included in the main analysis of maternal smoking effects (S4 Table). In GenR, 1,221 women (26%) reported smoking prior to pregnancy, including 423 who quit early in pregnancy (9%) and 798 (17%) who continued smoking through pregnancy. Corresponding percentages of pre-pregnancy smoking (41%) and continued smoking (30%)

were higher in BiB, whereas the proportion of women quitting in early pregnancy was similar to that observed in GenR (11%) (Table 1). Overall, 44% and 35% of partners smoked in GenR and BiB, respectively. In both cohorts, mothers who continued smoking during pregnancy were younger, shorter, less educated, and more often multiparous, and gave birth to infants of lower gestational age and birth weight (S5 Table). In BiB, mothers who continued smoking also had a lower BMI (S5 Table). Similar associations with these characteristics were observed for partner smoking status during pregnancy (S6 Table).

## Associations of maternal smoking with fetal growth

Differences in mean fetal size across gestation by maternal smoking during pregnancy are presented in Fig 1, with estimates from multivariable adjusted analyses shown in S7 Table. Overall, trajectories of fetal growth varied according to maternal smoking status ($p < 0.001$ for each fetal parameter). From the early second trimester through to term, fetuses of women who continued to smoke weighed less than those of non-smokers. More specifically, at 20 weeks, the predicted difference (95% CI) in mean EFW was −2.6 g (−5.1 to −0.1), and this absolute difference increased to −207 g (−231 to −182) at 40 weeks' gestation. A similar pattern was observed

**Table 1. Participant characteristics of the study population stratified by cohort.**

| Characteristic | GenR (n = 4,682) | BiB (n = 3,939) |
|---|---|---|
| Maternal age, years, mean (SD) | 31.3 (4.6) | 26.7 (6.0) |
| Smoking during pregnancy, percent (n) | | |
| Non-smoker | 73.9 (3,461) | 59.1 (2,328) |
| Pre-pregnancy smoker who quit before the second trimester | 9.0 (423) | 10.8 (427) |
| Pre-pregnancy smoker who continued through pregnancy | 17.0 (798) | 30.1 (1,184) |
| Smoking intensity in pre-pregnancy smokers who continued through pregnancy, percent (n) | | |
| Light | 40.3 (309) | 71.5 (2,328) |
| Moderate | 38.5 (295) | 19.0 (618) |
| Heavy | 21.3 (163) | 9.5 (309) |
| Missing | 3.9 (31) | 0 (0) |
| Partner smoking during pregnancy, percent (n) | | |
| No | 56.4 (2,373) | 64.8 (862) |
| Yes | 43.6 (1,833) | 35.2 (469) |
| Missing | 10.2 (476) | 66.2 (2,608) |
| Smoking intensity in partner's smoking during pregnancy, percent (n) | | |
| Light | 35.1 (633) | 18.3 (84) |
| Moderate | 16.8 (303) | 37.4 (172) |
| Heavy | 48.1 (868) | 44.3 (204) |
| Missing | 1.6 (29) | 1.9 (9) |
| Maternal rs1051730 genotype (number of T alleles), percent (n) | | |
| 0 | 45.3 (1,633) | 45.7 (1,337) |
| 1 | 43.0 (1,551) | 43.7 (1,277) |
| 2 | 11.7 (420) | 10.6 (309) |
| Missing | 23.0 (1,078) | 25.8 (1,016) |

For all smoking variables, percentages (n) of each category are given only for singleton births with no missing values, to facilitate comparison between the 2 cohorts.

BiB, Born in Bradford study; GenR, Generation R Study.

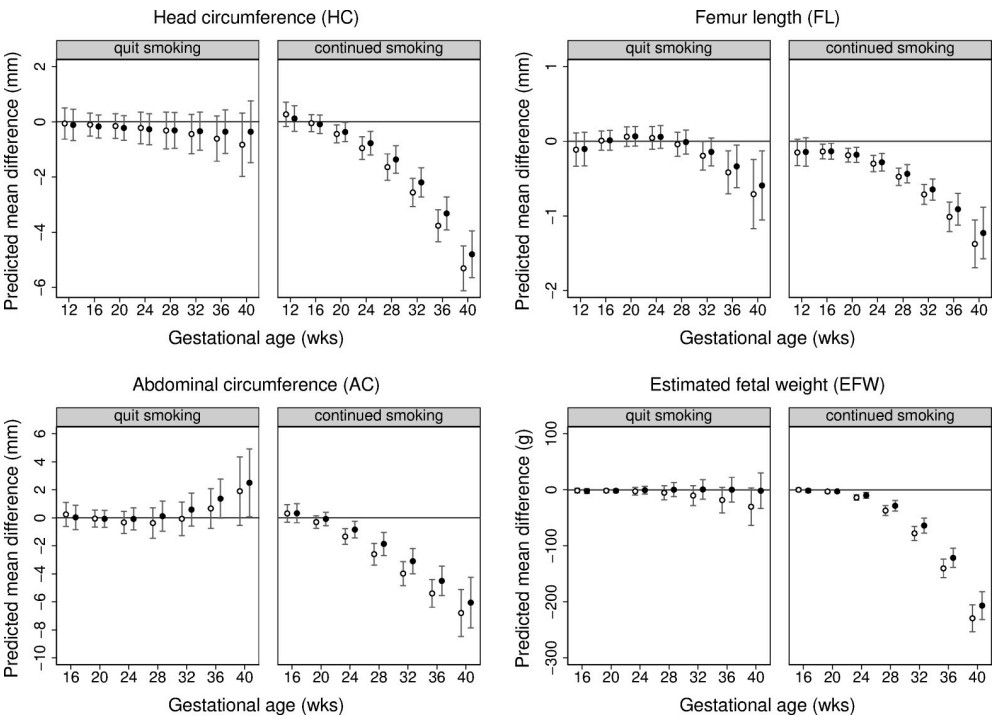

**Fig 1. Predicted differences in mean fetal size (with 95% CIs) across gestation comparing pre-pregnancy smokers who quit early in pregnancy and pre-pregnancy smokers who continued smoking through pregnancy with non-smokers (reference category).** Predicted differences in mean head circumference (mm), femur length (mm), abdominal circumference (mm), and estimated fetal weight (g) across gestation comparing pre-pregnancy smokers who quit in early pregnancy and those who continued smoking during pregnancy with non-smokers (reference category). Predicted mean differences (with 95% confidence intervals) in the pooled Generation R Study and Born in Bradford study cohort by analysis model: model 1 adjusting for cohort only (in white) and model 2 adjusting for cohort, infant sex, and maternal age, parity, height, body mass index, education level, and alcohol use during pregnancy (in black).

for HC, FL, and AC, with absolute differences in each fetal parameter widening with gestational age. Fetal FL reductions were observed from 16 weeks, and fetal HC and AC reductions were observed from 20 and 24 weeks, respectively. The EFW trajectory of women who quit smoking early in pregnancy was similar to that of non-smokers (Fig 1; S7 Table). However, quitters had a shorter FL and greater AC than non-smokers towards the end of pregnancy, though these differences were smaller in magnitude than those observed for continuing smokers.

Fetal size differences by smoking quantity in mothers who continued smoking during pregnancy are shown in Fig 2 and S8 Table. Dose–response associations with EFW and individual fetal parameters (HC, FL, and AC) were observed from early gestation through to birth ($p < 0.001$ for all fetal parameters). Compared to non-smokers, the difference (95% CI) in mean EFW for light, moderate, and heavy smoking in women who continued smoking was −144 g (−182 to −106), −215 g (−248 to −182), and −290 g (−334 to −247) at 40 weeks, respectively. As was the case for smoking status, associations of smoking quantity with fetal parameters appeared to be most pronounced for FL. Results for predicted differences as a proportion of the mean revealed a similar pattern as for mean differences in absolute original units: proportional differences in fetal parameters observed with maternal smoking status and smoking quantity followed an increasing pattern with increasing gestational age (S9 and S10 Tables).

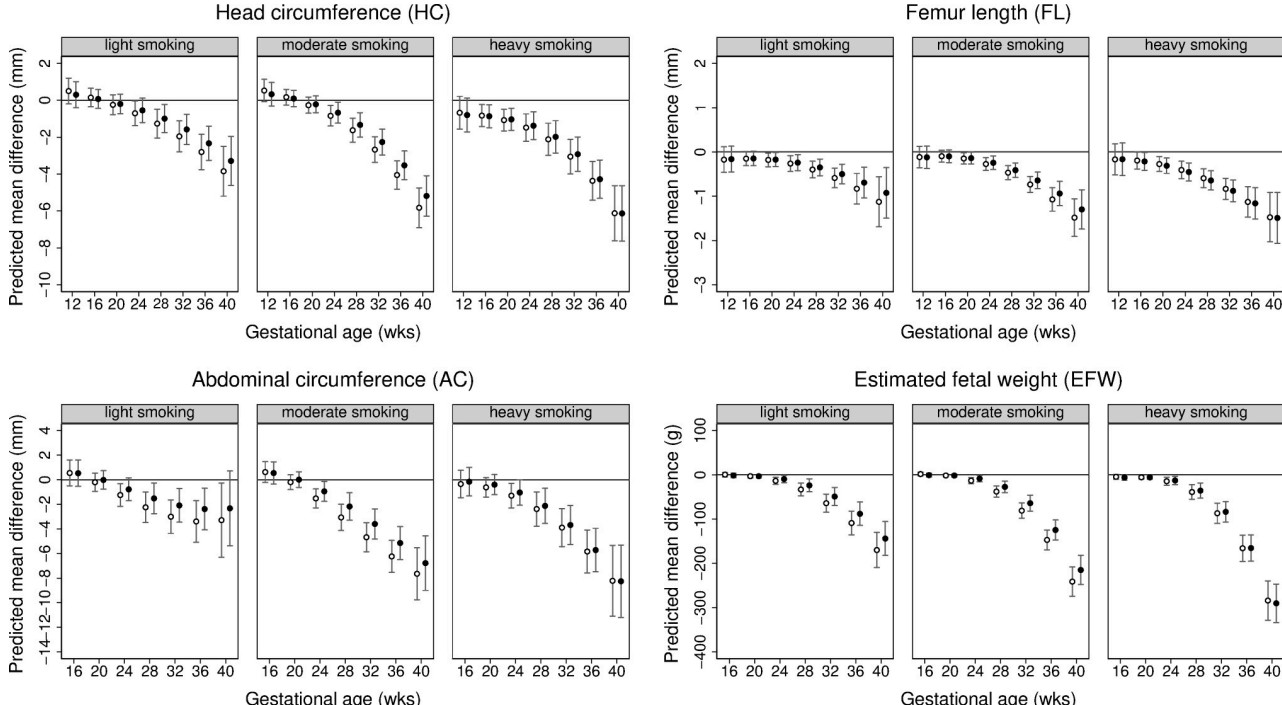

**Fig 2. Predicted differences in mean fetal size (with 95% CIs) across gestation comparing different categories of smoking quantity in pre-pregnancy smokers who continued smoking through pregnancy with non-smokers (reference category).** Predicted differences in mean head circumference (mm), femur length (mm), abdominal circumference (mm), and estimated fetal weight (g) across gestation comparing pre-pregnancy smokers who continued smoking through pregnancy by smoking intensity with non-smokers (reference category). Predicted mean differences (with 95% confidence intervals) in the pooled Generation R Study and Born in Bradford study cohort by analysis model: model 1 adjusting for cohort only (in white) and model 2 adjusting for cohort, infant sex, and maternal age, parity, height, body mass index, education level, and alcohol use during pregnancy (in black).

## Association of maternal rs1051730 genotype with smoking and fetal growth—MR analysis

Maternal rs1051730 genotype was associated with the ability to quit smoking among pre-pregnancy smokers (per T allele odds ratio [OR] for continuing versus quitting smoking during pregnancy = 1.10 [95% CI 0.96 to 1.25]). This association was mainly driven by GenR (OR = 1.17 [95% CI 0.96 to 1.43]) as rs1051730 was not clearly associated with the ability to quit smoking in BiB (OR = 1.03 [95% CI 0.86 to 1.24]) (S11 Table). In both cohorts, maternal rs1051730 genotype was associated with the number of cigarettes smoked in women who continued smoking during pregnancy (per T allele OR for heavy smoking versus light/moderate smoking = 1.24 [95% CI 1.04 to 1.47]), but not in those who quit smoking in early pregnancy (OR = 1.02 [95% CI 0.79 to 1.33]). The SNP was not associated with being a smoker (per T allele OR for being a pre-pregnancy smoker = 1.00 [95% CI 0.92 to 1.08]), confirming genome-wide association results that rs1051730 is not associated with smoking initiation [18].

Associations of maternal rs1051730 genotype with fetal growth differed by smoking status (Fig 3; S12 Table). Amongst women who continued to smoke across pregnancy, EFW growth differed by rs1051730 genotype ($p$ = 0.01). At 20 weeks' gestation, each additional T allele was associated with a lower EFW (−3.7 g [95% CI −7.0 to −0.4]), and this absolute reduction in fetal weight increased in magnitude with gestational age (at 40 weeks it was −45 g [95% CI −81 to −10]). A similar pattern of fetal growth restriction was observed for individual fetal

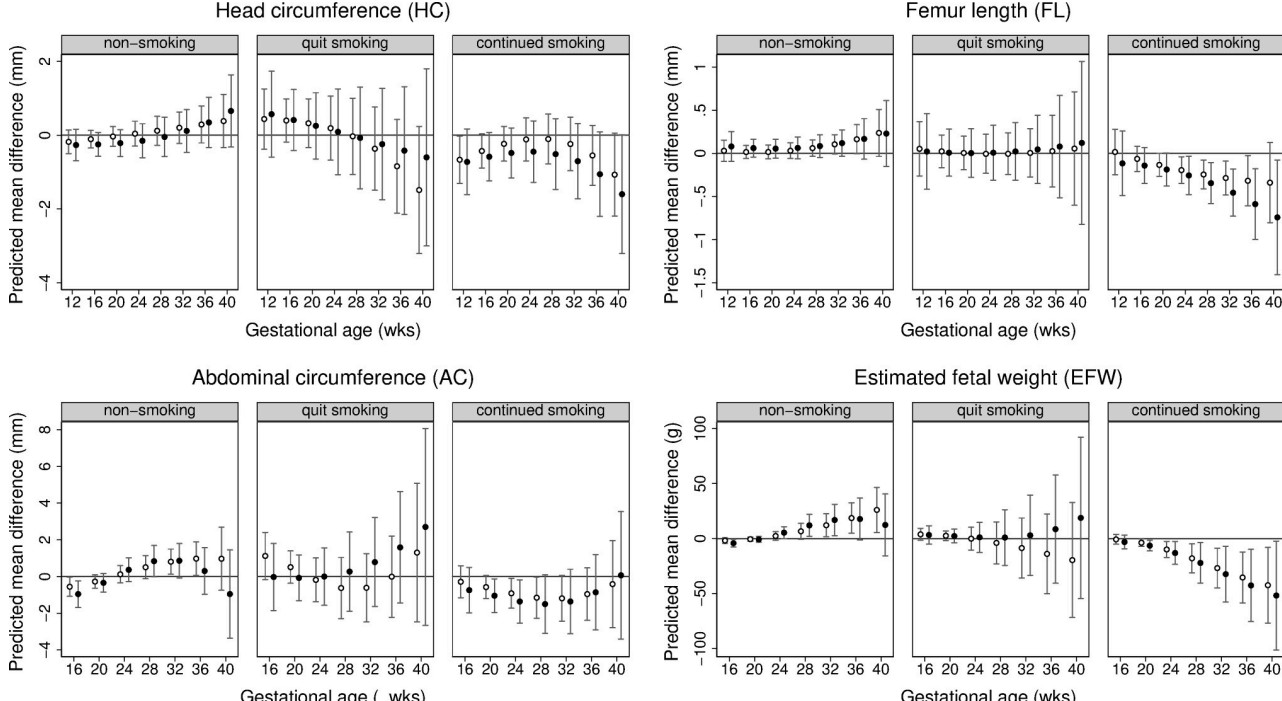

**Fig 3. Predicted differences in mean fetal size (with 95% CIs) across gestation per risk allele increase at rs1051730 in non-smokers, pre-pregnancy smokers who quit early in pregnancy, and pre-pregnancy smokers who continued smoking through pregnancy.** Predicted differences in mean head circumference (mm), femur length (mm), abdominal circumference (mm), and estimated fetal weight (g) per maternal rs1051730 T allele increase in non-smokers, pre-pregnancy smokers who quit smoking in early pregnancy, and pre-pregnancy smokers who continued smoking during pregnancy. Predicted mean differences (with 95% confidence intervals) by analysis model: model 1 adjusting for cohort (in white) and model 2 adjusting for cohort and fetal rs1051730 genotype (in black).

parameters (HC, FL, and AC) in these women, though differences in mean fetal AC were less precise and close to null in late gestation. In women classified as non-smokers, each additional T allele at rs1051730 was associated with a higher EFW between 32 weeks and 40 weeks of gestation ($p$ for overall difference in growth = 0.03), with similar patterns of associations for HC, AC, and FL. No association between maternal rs1051730 genotype and fetal growth was observed in women who quit smoking in early pregnancy, but these results need to be interpreted with caution because of the small number of individuals in this group. Proportional differences in mean fetal size with each additional T allele at rs1051730 followed a similar pattern as for the absolute differences observed (S13 Table). Effect estimates were not materially different after combining pre-pregnancy smokers who quit and those who continued smoking (S14 Table; S3 Fig). With the exception of maternal age, the SNP was not associated with confounders (S15 Table). Maternal age in continuing smokers increased with each additional T allele; there was no association with age in non-smokers and those who quit smoking in early pregnancy. Results were not altered with adjustment for maternal age (S4 and S5 Figs) or fetal rs1051730 genotype (Figs 3 and S3–S5).

## Association of maternal versus mother's partner smoking with fetal growth—Parental negative control analysis

In unadjusted analyses, partner smoking was associated with slower growth of all parameters (S6 Fig), but these associations were considerably weaker than those for maternal smoking and were only apparent later in pregnancy (from 32 weeks onwards). The partner smoking

associations were attenuated to the null after multivariable adjustment for confounders and mutual adjustment for maternal smoking (Figs 4 and S6; S16 and S17 Tables). Results for partner smoking quantity showed similar patterns (S7 Fig).

## Additional analyses

Effect estimates of the association between mother's and mother's partner's smoking status during pregnancy with fetal growth were not materially different in analyses restricted to those with complete covariate data (S8–S12 Figs). Associations were very similar in GenR and BiB, with no statistical evidence for any differences between the 2 cohorts ($p_{\text{interaction}} > 0.05$ for all analysis models; S7–S10 and S12–S17 Tables). Also, associations of maternal smoking with birth weight were consistent in direction and effect size with the fetal size differences observed during late gestation, and conditioning on gestational age did not strongly influence these associations (S18 Table).

## Discussion

In this study we found that infants of pre-pregnancy smokers who continued smoking during pregnancy had a reduced fetal size from early second trimester to term, and that this

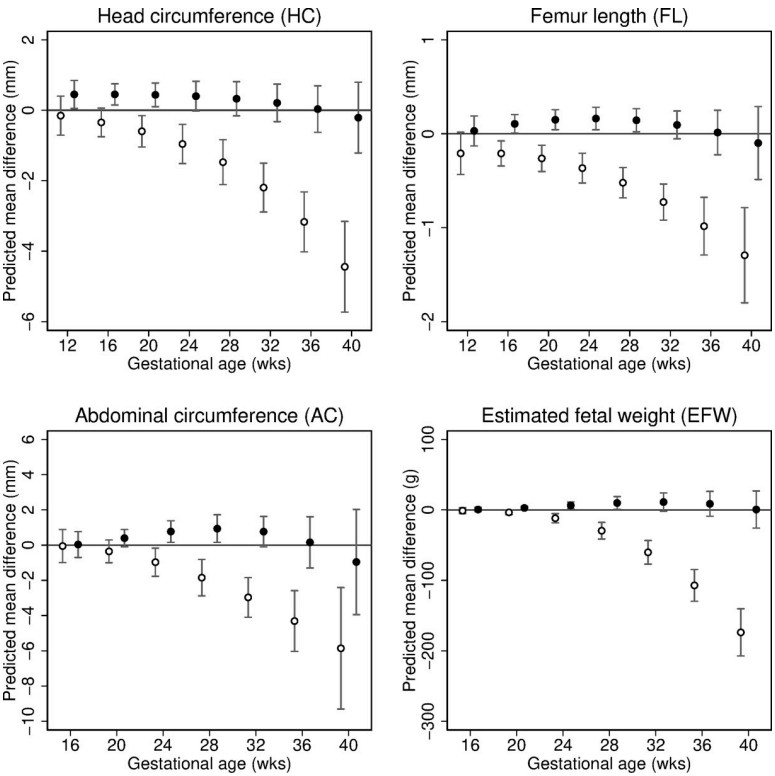

**Fig 4. Predicted differences in mean fetal size (with 95% CIs) across gestation comparing mothers and mother's partners who smoked to non-smokers (reference category).** Predicted differences in mean head circumference (mm), femur length (mm), abdominal circumference (mm), and estimated fetal weight (g) across gestation associated with maternal smoking (i.e., comparing maternal continued smoking through pregnancy with no maternal smoking during pregnancy [reference category]) and mother's partner's smoking (comparing partner smoking during pregnancy with no partner smoking during pregnancy [reference category]). Predicted mean differences associated with maternal smoking (in white) and partner smoking (in black) are adjusted for cohort, infant sex, parity (for maternal smoking only), and respective parental age, height, BMI, education level, and alcohol use during pregnancy, and mutually adjusted for the smoking behaviour of the other parent.

association followed a dose-dependent pattern (with reductions in fetal size being present in light smokers but of lower magnitude than in moderate or heavy smokers) and was independent of observed confounding factors. Furthermore, the lack of an equivalent association for partner's smoking with fetal growth and the results from MR, which showed evidence of a linear association of increasing cigarette consumption with fetal growth restriction, provide support that these associations represent an intrauterine effect and are not explained by unmeasured residual confounding factors. Pre-pregnancy smokers who gave up smoking before the second trimester had broadly similar fetal growth to non-smokers except for a shorter FL and greater AC between 36 and 40 weeks of gestation.

## Strengths and limitations

A strength of our study is the large population-based sample with detailed information on smoking (including dose–response data) and on potential confounders. We triangulated evidence from conventional multivariable regression analyses in mothers with evidence from parental negative control and genetic MR analyses. The consistent findings across these 3 methods—which have differing key sources of bias (residual confounding in the multivariable analyses, unanticipated causal effect of partner smoking in the negative control, and genetic horizontal pleiotropy in MR)—strengthen confidence in our conclusions [13]. Multiple sensitivity analyses suggested that the multivariable regression analyses were not biased by selection due to missing covariate data, and that stratified genetic analyses were not notably influenced by collider bias. An assumption of MR analysis is that the genetic instrument influences the outcome only through the exposure of interest. One theoretical source of violation of this assumption is via fetal genotype, but we found no difference in the maternal SNP–fetal growth associations with adjustment for fetal genotype. The positive association of maternal rs1051730 genotype with EFW between 32 and 40 weeks of gestation in non-smokers suggests that genetic pleiotropy may have resulted in violation of the assumption that the genetic instrument influences fetal growth only via maternal smoking. Previous MR studies using the rs1051730 genotype have demonstrated that smoking is causally related to a lower BMI in smokers and higher BMI, and waist and hip circumference, in non-smokers [27,28]. This raises the possibility that the association of the variant with offspring EFW in non-smokers is mediated through an increase in maternal body mass and that in smokers associations may be stronger (i.e., there may have been some masking of the association by a pleiotropic path through maternal BMI). Our MR results are consistent with those from a large study of 26,241 mother–infant pairs, which found a negative association of rs1051730 with birth weight in smokers but no strong evidence of an association in non-smokers (difference in mean birth weight per T allele = 5 g [95% CI −4 to 14]) [8].

The parental negative control study assumes that factors related to maternal smoking are similar to those related to their partner's smoking (i.e., that confounding is the same for the 'real' and 'negative control' exposure), which is plausible given that risk factors for smoking, such as low socioeconomic position, have not been shown to be sex-specific. This approach also assumes that it is not plausible that the negative control influences the outcome. Whilst partner smoking, and that of others whom the mother is exposed to, could have an intrauterine effect via passive smoking, we would expect this effect to be considerably smaller than what is seen with maternal smoking. The fact that we see in our analyses presented here a maternal-specific association with fetal growth parameters, as has been shown for birth weight [9], supports an intrauterine mechanism that is not explained by shared familial confounding.

In both GenR and BiB, pregnancies were dated using fetal biometry using CRL and biparietal diameter. Although this approach is superior to the use of women's self-report of last

menstrual period, it assumes no variation in fetal size at the time of dating, which may have led to an underestimation of smoking associations. Moreover, as there is some evidence of maternal smoking being associated with reduced first trimester CRL [11], systematic underestimation of gestational age in women who smoke during pregnancy could have further limited our ability to detect (stronger) associations. Since we modelled longitudinal fetal growth, and since ultrasound fetal anthropometrics tend to correlate less with dating measures as pregnancy progresses, we expect this to have resulted in an underestimation of fetal size differences only during early gestation. We further note that associations of maternal smoking with fetal AC attenuated towards term, most likely because of limited power as AC was not measured at birth in GenR. Information about smoking during pregnancy was collected by questionnaire. Several studies using cotinine as a biochemical measure of smoking exposure have demonstrated underreporting of smoking among pregnant women [29,30]. It is therefore likely that not all women in our study admitted smoking and when reporting smoking may have systematically underestimated the number of cigarettes smoked or falsely reported having quit. In the multi-variable regression analyses, this misreporting could have resulted in associations with whether mothers smoked or not being biased towards the null, and associations with light/moderate smoking and smoking cessation being biased away from the null. In MR analyses, this misclassi-fication would be expected to potentially bias results for non-smokers and quitters towards the results for smokers (i.e., as the group of non-smokers and quitters may have included some smokers), but not to bias causal effect estimates for smoking quantity within smokers.

Partners can be difficult to recruit to pregnancy cohorts, and data on behaviours like smoking is sometimes collected from the pregnant woman (as was the case for most partners in GenR) or there is a smaller proportion of partners with data (as was the case for BiB) [31]. The fact that we found consistent results for partner smoking in GenR, where information on partner smoking was based predominantly on maternal report (which may be biased towards the maternal association), and BiB, in which only a small proportion of partners responded (where there may be selection bias), suggests that the null findings for partner smoking are robust. Finally, as this study included women of white European origin only, results may not necessarily generalise to other populations.

## Comparison with other studies

Our results are consistent with those from a systematic review, which reported lower third trimester fetal size and second trimester reductions in some fetal parameters in women who continued smoking during pregnancy [12]. That review, however, could not assess fetal growth trajectories across pregnancy from early gestation to term, or explore dose-dependent patterns or the impact of quitting in detail, and it did not include methods we used to explore causality. Our MR estimates of EFW reductions of 37 g (95% CI 14 to 60) and 45 g (95% CI 10 to 81) per genetically determined 1-cigarette-per-day increase at 36 and 40 weeks' gestation in women who continued smoking beyond the first trimester are broadly consistent with the 24-g (95% CI 3 to 45) reduction in birth weight reported in a previous MR analysis [8]. Our genetic results are also comparable with those from a randomized smoking cessation trial showing a reduction of approximately 5.4 cigarettes per day and a 92-g heavier birth weight in the intervention compared to the control group (giving an average increase of 17.9 g per 1 less cigarette) [5]. Of note, rs1051730 genotype is an instrument for both smoking quantity (in current smokers, i.e., pre-pregnancy smokers who continue smoking during pregnancy) and smoking cessation (in pre-pregnancy smokers), 2 independent traits that are strongly correlated. As we found similar associations in analyses including all pre-pregnancy smokers, our results support the importance of smoking reduction and cessation to prevent fetal growth restriction. The

absence of an independent association of mother's partner's smoking with fetal growth provides additional support for maternal smoking influencing early fetal growth through an intra-uterine mechanism and is consistent with a previous negative control study investigating parental smoking associations with birth weight [9].

Different fetal parameters have different peak periods of growth during gestation as reflected by the shapes of their growth trajectories, i.e., linear growth of HC and FL is highest during early pregnancy and starts to level off at approximately 20 weeks of gestation, while the peak growth of AC is observed between approximately 20 and 32 weeks (reflecting the predominant accumulation of fat and lean tissue during the third trimester) [32]. These differences in fetal growth patterns may affect the timing at which associations with each fetal parameter can be observed. Differences in smoking effects on fetal growth parameters may also reflect developmental plasticity and the 'protection' of some organs and tissues over others. In our study, associations of maternal smoking with fetal growth were first noted for FL, with differences in FL being detectable from as early as 16 weeks' gestation. Compared to non-smokers, pre-pregnancy smokers who quit smoking before the second trimester had a shorter fetal FL at 36–40 weeks' gestation, whereas no differences in size were observed for other growth parameters. If the toxic effects of smoking affected the growth of different tissues and organs similarly and the main cause of differences between them was related to timing of peak growth, we would have expected similar patterns between FL and HC, which is not what we find.

A predominant association of maternal smoking with FL has been reported before by other studies including GenR [20,33–35] and animal experiments [36], and for many years, leg length has been proposed to be the most sensitive growth parameter to adverse early-life exposures [37]. A Swedish register-based study found that infants of women who quit smoking during pregnancy had a similar birth weight and HC to infants of non-smokers, but a shorter crown–heel length [38]. Since birth crown–heel length is a measure of linear growth that is closely correlated to FL [39], these data are consistent with our findings. Taking our findings together with those from human and animal experimental studies, it appears that fetal skeletal linear growth is particularly susceptible to maternal cigarette smoking. It seems plausible that in the presence of reduced nutrient and oxygen supply to the fetus, as a result of smoking [40], key organs, such as the brain and liver, are preserved as much as possible at the expense of skeletal growth. We further observed a greater fetal AC towards the end of gestation in women who quit smoking early in pregnancy compared to non-smokers. Smoking cessation is often accompanied by weight gain and changes in central fat tissue [41,42], most likely through an increased appetite and lower basal metabolic rate upon nicotine withdrawal. This could potentially explain the larger fetal AC found among quitters, but because of the small number of quitters in this study, we were unable to explore the causal nature of this association further in MR analysis.

## Implications

Harmful effects of maternal smoking during pregnancy are well known. However, many women who smoke find it difficult to quit before or during pregnancy [43], as evidenced by the relatively high rates of smoking prior to pregnancy (26% and 41%) and low quit rates in early pregnancy (35% and 27% of pre-pregnancy smokers) in the 2 contemporary cohorts of white European women in our study. Our study is the first to our knowledge to provide robust support for an association between the number of cigarettes smoked and fetal size from as early as 20 weeks' gestation in women who continue smoking after the first trimester. There are 2 key implications of these findings. First, even at low doses, cigarette smoking adversely affects fetal growth, and continued efforts to prevent women of reproductive age from smoking, and to support those who do take it up to quit, are paramount. In this context, our findings

reinforce existing recommendations [44–46] that encourage strategies to reduce initiation, improve detection, and promote cessation of tobacco use including cessation counselling as part of preconception health services and antenatal care. Simultaneously, the absence of major fetal growth deficits observed with smoking cessation early in pregnancy should comfort women with unplanned pregnancies who are able to quit smoking early after realising they are pregnant, but who may have experienced anxiety because of smoking in that early period of pregnancy. Second, for women who do not manage to quit smoking before or during pregnancy, smoking reduction will have some benefit in a linear fashion, i.e., the greater the reduction in smoking, the smaller the likelihood of fetal growth restriction; therefore, support to lower the number of cigarettes smoked in these women should be promoted. Currently, few guidelines suggest this, perhaps because of fear that it may be interpreted as condoning light smoking in pregnancy. We believe our results provide sufficient evidence to update guidelines that currently promote only quitting, not reducing, smoking: For women who find it too difficult to quit smoking before or during pregnancy, support to reduce the amount they smoke should be provided. We acknowledge that fetal growth restriction, despite being a predictor of neonatal morbidity and mortality, is only one of the adverse outcomes associated with pregnancy smoking, and it is unknown whether lowering cigarette consumption in those who cannot quit will be beneficial for other perinatal outcomes. Finally, our findings should stimulate more research to further explore the effectiveness and safety of currently available interventions that might reduce smoking in pregnancy [47]. The potential effect of electronic cigarettes and other smokeless tobacco products, which women may use as a means to quit smoking or as potentially safer alternatives to cigarette smoking, on fetal growth and perinatal outcomes requires investigation. However, as these alternatives have only recently become available, we did not have information on their use in either of these cohorts, and it will take some time before other cohorts with sufficient information on these will be available for analyses.

## Conclusions

By triangulating findings from different analytical approaches, this study provides strong support for a dose-dependent effect of cigarette smoking on fetal growth from the second trimester onwards in women who continue smoking during pregnancy, while only minor deficits in fetal growth are seen in women who quit smoking early in pregnancy. Collectively, these findings reinforce the importance of smoking cessation counselling in preconception health services and antenatal care. They demonstrate the importance of quitting smoking early in pregnancy, which is knowledge that could also help reduce anxiety in women who quit after learning they are pregnant. Lastly, our findings indicate that support and advice to reduce smoking during pregnancy in women who find it impossible to quit has some benefit in lowering the risk of fetal growth restriction.

## Supporting information

**S1 Fig. Flow chart of the study population.**
(TIF)

**S2 Fig. Best-fitting growth trajectories for each fetal parameter identified by multilevel fractional polynomial models in GenR and BiB.** Average growth trajectories of fetal HC, FL, AC, and EFW from 12–16 to 40 weeks' gestation predicted by best-fitting multilevel fractional polynomial models in GenR and BiB.
(TIF)

**S3 Fig. Predicted differences in mean fetal size (with 95% CIs) across gestation per risk allele increase at rs1051730 in pre-pregnancy smokers and non-smokers.** Predicted differences in mean HC (mm), FL (mm), AC (mm), and EFW (g) across gestation per maternal rs1051730 T allele increase in pre-pregnancy smokers and non-smokers. Predicted mean differences (with 95% confidence intervals) in the pooled GenR and BiB cohort by analysis model: model 1 adjusting for cohort (in white) and model 2 adjusting for cohort and fetal rs1051730 genotype (in black).
(TIF)

**S4 Fig. Predicted differences in mean fetal size (with 95% CIs) across gestation per risk allele increase at rs1051730 in non-smokers, pre-pregnancy smokers who quit in early pregnancy, and pre-pregnancy smokers who continued smoking during pregnancy—Analysis with adjustment for maternal age.** Predicted differences in mean HC (mm), FL (mm), AC (mm), and EFW (g) across gestation per maternal rs1051730 T allele increase in non-smokers, pre-pregnancy smokers who quit smoking before the second trimester, and pre-pregnancy smokers who continued smoking during pregnancy. Predicted mean differences (with 95% confidence intervals) in the pooled GenR and BiB cohort by analysis model: model 1 adjusting for cohort and maternal age (in white) and model 2 adjusting for cohort, maternal age, and fetal rs1051730 genotype (in black).
(TIF)

**S5 Fig. Predicted differences in mean fetal size (with 95% CIs) across gestation per risk allele increase at rs1051730 in pre-pregnancy smokers and non-smokers—Analysis with adjustment for maternal age.** Predicted differences in mean HC (mm), FL (mm), AC (mm), and EFW (g) across gestation per maternal rs1051730 T allele increase in pre-pregnancy smokers and non-smokers. Predicted mean differences (with 95% confidence intervals) in the pooled GenR and BiB cohort by analysis model: model 1 adjusting for cohort and maternal age (in white) and model 2 adjusting for cohort, maternal age, and fetal rs1051730 genotype (in black).
(TIF)

**S6 Fig. Predicted differences in mean fetal size (with 95% CIs) across gestation comparing mothers' partners who smoked with non-smoking partners (reference category).** Predicted differences in mean HC (mm), FL (mm), AC (mm), and EFW (g) across gestation comparing partners who smoked during pregnancy to those who did not smoke during pregnancy (reference category). Predicted mean differences (with 95% confidence intervals) in the pooled GenR and BiB cohort by analysis model: model 1 adjusting for cohort only (in white); model 2 adjusting for cohort, infant sex, and partner age, height, body mass index, education level, and alcohol use during pregnancy (in grey); and model 3 adjusting for cohort; infant sex; partner age, height, body mass index, education level, and alcohol use during pregnancy; and smoking during pregnancy (in black).
(TIF)

**S7 Fig. Predicted differences in mean fetal size (with 95% CIs) across gestation comparing different categories of smoking intensity in mothers' partners who smoked with non-smoking partners (reference category).** Predicted differences in mean HC (mm), FL (mm), AC (mm), and EFW (g) across gestation comparing partners who smoked during pregnancy by smoking intensity with those who did not smoke during pregnancy (reference category). Predicted mean differences (with 95% confidence intervals) in the pooled GenR and BiB cohort by analysis model: model 1 adjusting for cohort only (in white); model 2 adjusting for cohort, infant sex, and partner age, height, body mass index, education level, and alcohol use

during pregnancy (in grey); and model 3 adjusting for cohort; infant sex; partner age, height, body mass index, education level, and alcohol use during pregnancy; and maternal smoking during pregnancy (in black).
(TIF)

**S8 Fig. Predicted differences in mean fetal size (with 95% CIs) across gestation comparing pre-pregnancy smokers who quit in early pregnancy and pre-pregnancy smokers who continued smoking through pregnancy with non-smokers (reference category)—Complete case analysis.** Predicted differences in mean HC (mm), FL (mm), AC (mm), and EFW (g) across gestation comparing pre-pregnancy smokers who quit smoking in early pregnancy and those who continued smoking during pregnancy with non-smokers (reference category)—complete case analysis. Predicted mean differences (with 95% confidence intervals) in the pooled GenR and BiB cohort by analysis model: model 1 adjusting for cohort only (in white) and model 2 adjusting for cohort, infant sex, and maternal age, parity, height, body mass index, education level, and alcohol use during pregnancy (in black).
(TIF)

**S9 Fig. Predicted differences in mean fetal size (with 95% CIs) across gestation comparing different categories of smoking intensity in pre-pregnancy smokers who continued smoking through pregnancy with non-smokers (reference category)—Complete case analysis.** Predicted differences in mean HC (mm), FL (mm), AC (mm), and EFW (g) across gestation comparing pre-pregnancy smokers who continued smoking through pregnancy by smoking intensity with non-smokers (reference category)—complete case analysis. Predicted mean differences (with 95% confidence intervals) in the pooled GenR and BiB cohort by analysis model: model 1 adjusting for cohort only (in white) and model 2 adjusting for cohort, infant sex, and maternal age, parity, height, body mass index, education level, and alcohol use during pregnancy (in black).
(TIF)

**S10 Fig. Predicted differences in mean fetal size (with 95% CIs) across gestation comparing mothers' partners who smoked with non-smoking partners (reference category)—Complete case analysis.** Predicted differences in mean HC (mm), FL (mm), AC (mm), and EFW (g) across gestation comparing partners who smoked during pregnancy with those who did not smoke during pregnancy (reference category)—complete case analysis. Predicted mean differences (with 95% confidence intervals) in the pooled GenR and BiB cohort by analysis model: model 1 adjusting for cohort only (in white); model 2 adjusting for cohort, infant sex, and partner age, height, body mass index, education level, and alcohol use during pregnancy (in grey); and model 3 adjusting for cohort; infant sex; partner age, height, body mass index, education level, and alcohol use during pregnancy; and maternal smoking during pregnancy (in black).
(TIF)

**S11 Fig. Predicted differences in mean fetal size (with 95% CIs) across gestation comparing different categories of smoking intensity in mothers' partners who smoked with non-smoking partners (reference category)—Complete case analysis.** Predicted differences in mean HC (mm), FL (mm), AC (mm), and EFW (g) across gestation comparing partners who smoked during pregnancy by smoking intensity with those who did not smoke during pregnancy (reference category)—complete case analysis. Predicted mean differences (with 95% confidence intervals) in the pooled GenR and BiB cohort by analysis model: model 1 adjusting for cohort only (in white); model 2 adjusting for cohort, infant sex, and partner age, height, body mass index, education level, and alcohol use during pregnancy (in grey); and model 3 adjusting for cohort; infant sex; partner age, height, body mass index, education level, and

alcohol use during pregnancy; and maternal smoking during pregnancy (in black).
(TIF)

**S12 Fig. Predicted differences in mean fetal size (with 95% CIs) across gestation comparing mothers and mothers' partners who smoked to non-smokers (reference category)—Complete case analysis.** Predicted differences in mean HC (mm), FL (mm), AC (mm), and EFW (g) across gestation associated with maternal smoking (i.e., comparing maternal continued smoking through pregnancy with no maternal smoking during pregnancy [reference category]) and mother's partner's smoking (comparing partner smoking during pregnancy with no partner smoking during pregnancy [reference category])—complete case analysis. Predicted mean differences associated with maternal smoking (in white) and partner smoking (in black) are adjusted for cohort, infant sex, parity (for maternal smoking only), and respective parental age, height, BMI, education level, and alcohol use during pregnancy, and mutually adjusted for the smoking behaviour of the other parent.
(TIF)

**S1 Table. Genotype quality control measures in GenR and BiB.**
(DOCX)

**S2 Table. Descriptive statistics of the repeat ultrasound and birth anthropometric measurements in GenR and BiB.**
(DOCX)

**S3 Table. Comparison of observed means of fetal size with those predicted by the multilevel fractional polynomial model in GenR and BiB.**
(DOCX)

**S4 Table. Participant characteristics by availability of maternal rs1051730 genotype and partner smoking data.**
(DOCX)

**S5 Table. Participant characteristics by maternal smoking status during pregnancy.**
(DOCX)

**S6 Table. Participant characteristics by partner smoking status during pregnancy.**
(DOCX)

**S7 Table. Maternal smoking during pregnancy and predicted differences in mean fetal size (with 95% CIs) across gestation, overall and stratified by cohort.**
(DOCX)

**S8 Table. Maternal smoking intensity in continuing smokers and predicted differences in mean fetal size (with 95% CIs) across gestation, overall and stratified by cohort.**
(DOCX)

**S9 Table. Maternal smoking during pregnancy and predicted differences in mean fetal size across gestation as a proportion of the mean, overall and stratified by cohort.**
(DOCX)

**S10 Table. Maternal smoking intensity in continuing smokers and predicted differences in mean fetal size across gestation as a proportion of the mean, overall and stratified by cohort.**
(DOCX)

**S11 Table. Association of maternal rs1051730 genotype with maternal smoking variables.**
(DOCX)

**S12 Table. Predicted differences in mean fetal size (with 95% CIs) across gestation per risk allele increase at rs1051730 in non-smokers, pre-pregnancy smokers who quit in early pregnancy, and pre-pregnancy smokers who continued smoking through pregnancy, overall and stratified by cohort.**
(DOCX)

**S13 Table. Predicted differences in mean fetal size across gestation as a proportion of the mean per risk allele increase at rs1051730 in non-smokers, pre-pregnancy smokers who quit in early pregnancy, and pre-pregnancy smokers who continued smoking through pregnancy, overall and stratified by cohort.**
(DOCX)

**S14 Table. Predicted differences in mean fetal size (with 95% CIs) across gestation per risk allele increase at rs1051730 in pre-pregnancy smokers and non-smokers, overall and stratified by cohort.**
(DOCX)

**S15 Table. Association of maternal rs1051730 genotype with potential confounders.**
(DOCX)

**S16 Table. Parental smoking during pregnancy and predicted differences in mean fetal size (with 95% CIs) across gestation, overall and stratified by cohort.**
(DOCX)

**S17 Table. Parental smoking during pregnancy and predicted differences in mean fetal size across gestation as a proportion of the mean, overall and stratified by cohort.**
(DOCX)

**S18 Table. Predicted differences in mean infant birth weight in grams (with 95% CIs) associated with maternal smoking during pregnancy and smoking intensity in continuing smokers.**
(DOCX)

**S1 Text. Supplemental methods.**
(DOCX)

**S2 Text. Analysis plan.**
(DOCX)

**S3 Text. STROBE checklist.**
(DOCX)

## Acknowledgments

BiB is only possible because of the enthusiasm and commitment of the children and parents. We are grateful to all the participants, practitioners, and researchers who have made BiB happen.

GenR is conducted by Erasmus University Medical Center Rotterdam, in close collaboration with the School of Law and Faculty of Social Sciences of the Erasmus University Rotterdam; the Municipal Health Service for Rotterdam area, Rotterdam; the Rotterdam Homecare Foundation Rotterdam; and the Stichting Trombosedienst & Artsenlaboratorium Rijnmond

(STAR-MDC), Rotterdam. We gratefully acknowledge the contribution of participating children and mothers, parents, general practitioners, hospitals, midwives, and pharmacies in Rotterdam. The generation and management of genome-wide association study (GWAS) genotype data was performed by the Human Genotyping Facility of the Genetic Laboratory of the Department of Internal Medicine, Erasmus MC, Rotterdam, the Netherlands. We thank Pascal Arp, Mila Jhamai, Marijn Verkerk, Lizbeth Herrera, and Marjolein Peters for their help in the creation, management, and quality control of the GWAS database, and Karol Estrada and Carolina Medina-Gomez for their support in the creation and analysis of imputed data. We would like to thank Anis Abuseiris, Karol Estrada, Dr. Tobias A. Knoch, and Rob de Graaf as well as their institution, Biophysical Genomics, Erasmus MC, Rotterdam, the Netherlands, for their help in creating GRIMP, BigGRID, MediGRID, and Services@MediGRID/D-Grid for access to their grid computing resources.

## Author Contributions

**Conceptualization:** Judith S. Brand, Romy Gaillard, John Wright, Kate Tilling, Deborah A. Lawlor.

**Data curation:** Romy Gaillard, Jane West, Rosemary R. C. McEachan, John Wright, Ellis Voerman, Janine F. Felix.

**Formal analysis:** Judith S. Brand.

**Funding acquisition:** Jane West, Rosemary R. C. McEachan, John Wright, Janine F. Felix, Deborah A. Lawlor.

**Methodology:** Judith S. Brand, Kate Tilling, Deborah A. Lawlor.

**Writing – original draft:** Judith S. Brand, Deborah A. Lawlor.

**Writing – review & editing:** Romy Gaillard, Jane West, Rosemary R. C. McEachan, John Wright, Ellis Voerman, Janine F. Felix, Kate Tilling.

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
