## [Decision Letter · Decision Letter 0]

13 Aug 2019

Dear Dr. Brand,

Thank you very much for submitting your manuscript "Effects of maternal quitting, reducing and continuing smoking during pregnancy on fetal growth trajectories: findings from Mendelian randomization and parental negative control studies" (PMEDICINE-D-19-01958) for consideration at PLOS Medicine. 

[LINK]

In light of these reviews, I am afraid that we will not be able to accept the manuscript for publication in the journal in its current form, but we would like to consider a revised version that addresses the reviewers' and editors' comments. Obviously we cannot make any decision about publication until we have seen the revised manuscript and your response, and we plan to seek re-review by one or more of the reviewers. 

We expect to receive your revised manuscript by Sep 03 2019 11:59PM. Please email us (plosmedicine@plos.org) if you have any questions or concerns.

We look forward to receiving your revised manuscript. 

Sincerely,

Clare Stone

Acting Editor-in-Chief

PLOS Medicine

plosmedicine.org

Academic Editor comments:

There are very good suggestions from the reviewers; those from reviewers 1 and 2 about adding more discussion on the novelty and practical implications of the findings will further clarify what this work adds to the available evidence.

- on data, it sounds like there are restrictions (i.e. "without restriction" is incorrect; presumably to do with the ethics approval, which could be mentioned briefly under data).

- I'd remove "Effects of ..." from the start of the title (also "... and fetal growth ..."). Please use association of as this is not a trial.

- In the abstract, I'd be interested to know some demographic details of cohort participants and the dates of inclusion; please state briefly the limitations of the study as the final sentence of the ‘Methods and Findings’ section of the abstract; please use the past tense in the conclusions section of the abstract (was oserved). Please add p values where 95%Cis are given. 

- Did your study have a prospective protocol or analysis plan? Please state this (either way) early in the Methods section.

c) In either case, changes in the analysis—including those made in response to peer review comments—should be identified as such in the Methods section of the paper, with rationale.

- "We found ..." and the findings in the past tense would be better in the first paragraph of the discussion

Please avoid causal language – eg line 346 “ are causal intrauterine effects”; line 340 – maybe have demonstrated an association that infants; robust evidence (line 470) and so on.

The STROBE looks to be in an unusual format. Please use the standard checklist and include sections and paragraphs. 

Comments from the reviewers:

Reviewer #1: This is a well-written epidemiological analysis utilising two population-based European pregnancy cohorts (Generation R Study and Born in Bradford Study) and several approaches in an attempt to provide greater nuanced evidence on the association between maternal smoking during pregnancy and fetal growth. While lacking in size compared to the 2017 Abraham M et al systematic review (A systematic review of maternal smoking during pregnancy and fetal measurements with meta-analysis), which also included results from the Generation R study, this work does attempt to address issues related to causal inference and fetal growth (over time).

Major comment:

Given the recent systematic review and other work that has examined maternal smoking and adverse reproductive outcomes, I believe much more effort needs to be made in highlighting the specific additional clinical practice and/or health policy implications of this manuscript. At the moment, the manuscript appears to reinforce the existing clinical/public health messaging and in my view this was not a point of contention. Although, not the focus of this manuscript, the association between E-cigarette use during pregnancy and reproductive health outcomes is largely unknown and the authors should consider if their current work dwells inordinately on the finer points of maternal smoking and fetal growth association ("circular epidemiology") rather than pursue new avenues of research in this area. 

In view of the above comment, the manuscript could be strengthened by a more detailed discussion on how the findings from this study may/may not infuence existing NHS advice/statements targeted to women who are pregnant. Below is an example of some of the current NHS advice/statements targeted to women who are pregnant:

No matter what stage you're at in your pregnancy, it's never too late to stop smoking.

Any woman referred for specialist advice to quit will get it.

Any expectant mum who smokes or has recently quit will be offered specialist support throughout pregnancy, including as late as 36 weeks into the baby's development.

If your partner or anyone else who lives with you smokes, their smoke can affect you and the baby both before and after birth. 

https://www.nhs.uk/smokefree/why-quit/smoking-in-pregnancy

https://www.england.nhs.uk/2019/04/quit-smoking-advice-for-pregnant-women-in-nhs-drive-to-prevent-stillbirths/

https://www.nhs.uk/conditions/pregnancy-and-baby/smoking-pregnant/

Reviewer #2: Overall: The use of two population-based pregnancy cohorts to assess the impact of smoking behaviors on low birth weight is an important contribution to the literature, and an opportunity to again emphasize the importance of tobacco prevention and control in reproductive age women (and men).

Specific Comments: 

Lines 64-65. The U.S. Surgeon General's Reports have discussed the subject of cigarette smoking's causal link with low birth weight in many volumes. For example, the 2004 Surgeon General's report concluded that "the evidence is sufficient to infer a causal relationship between maternal active smoking and fetal growth restriction and low birth weight (USDHHS 2004, p. 28). This conclusion was based on numerous studies with consistent evidence of a dose-response relationship and confirmed by studies that use biomarkers of exposure to tobacco smoke, among others. Also, the 2010 Surgeon General's report (The Biology and Behavioral Basis for Smoking-Attributable Disease) includes a lengthy discussion of the biological basis for smoking's link to low birth weight. I suggest the authors reference one or more of the Surgeon General's reports, or otherwise make clearer that the causal relationship between cigarette smoking and low birth weight is long-standing and very strong, and the evidence for it is multi-faceted.

Line 106: Here and elsewhere, I would strongly advise against using the term "habit" to refer to smoking - given the highly addictive nature of tobacco products. The term "behavior" or "use" is preferred.

Lines 122-124. Given that women had several assessments, how was CPD calculated? Was it averaged over the several assessments? Also, partners' smoking is included in table 1 - were the same parameters (in terms of CPD) used?

Table 1 - is there a typo? The numbers for "partner smoking during pregnancy" for the BiB cohort (64.8%; 35.2%; 66.2%) don't seem to make sense.

Lines 385-388. Very important to acknowledge that many studies show that pregnant women often do not acknowledge or accurately report their smoking status. 

In a few places the authors make reference to smoking reduction (rather than complete smoking cessation). E.G., Lines 58 and 420. As regards low birth weight, the authors make a good case that while complete cessation is best, smoking reduction is also likely to be helpful, and they do caveat the "reduction" sections by noting that this is "beneficial in terms of fetal growth." However, I suggest adding a sentence indicating that low birth weight is just one of many adverse effects of smoking on fetal outcomes - and we do not yet know whether smoking reduction is helpful - to emphasize the point that low birth weight is not the only parameter of interest.

Lines 473-474: Re: "the importance of good smoking cessation advice in preconception health services and antenatal care." This sentence is inadequate to describe what is needed to address the long-standing problem of pregnant women's tobacco use. Although not the primary focus of this paper, I would suggest the authors consider expanding this section. Some useful additional points may include: 

1) rates of smoking during pregnancy are reflective of rates of smoking among reproductive age women. Therefore, doing more to address smoking (and other tobacco product use) in this population - and among women and men in this age group generally - is imperative. 

2) providing cessation counseling as an integral part of preconception health services and antenatal care is critical - but this addresses only the woman herself, and not the family and larger community/policy context in which tobacco use among pregnant women occurs. Strong, evidence-based tools are available to prevent and control smoking among both women and men. Good references for this information include:

WHO recommendations for the prevention and management of tobacco use and second-hand smoke exposure in pregnancy. Available here: https://www.who.int/tobacco/publications/pregnancy/guidelinestobaccosmokeexposure/en/

In addition, to providing guidance on effective screening and both psychosocial and pharmacological interventions, the guidelines explain that the responsibility for quitting does not "rest" entirely with the woman herself. There is an obligation for the family, community, and society to create an environment that support her doing so. For example, p. 37: 

"…all population-level policies and interventions for comprehensive tobacco control that are proven to be effective for the general population, would also help protect the health of pregnant women as well. These policies create an enabling environment which is promotive of non-use of tobacco and enable and empower women to be able to implement their own choices."

Effective population-level policies and interventions are encompassed by the WHO Framework Convention on Tobacco Control (WHO FCTC) and are also described in the U.S. National Cancer Institute Tobacco Control Monograph 21, The Economics of Tobacco and Tobacco Control, published in 2016, among many other places. Available here: https://cancercontrol.cancer.gov/brp/tcrb/monographs/21/index.html

Reviewer #3: I confine my remarks to statistical aspects of this paper. These were very well done indeed - I almost clicked "accept" - but I do have a couple suggestions on the figures.

First, make it clear in the header that "nonsmokers" are the reference group. More substantively, I think a line plot with confidence bands might be better, although the difference will be minor. Finally, I'm not sure about including model 1 and 2. Are both really needed? The estimates are close. I'd go with model 2.

But, overall, excellent work

Peter Flom

[LINK]

---

## [Editor Report · Decision Letter 1]

4 Oct 2019

Dear Dr. Brand,

Thank you very much for re-submitting your manuscript "Effects of maternal quitting, reducing and continuing smoking during pregnancy on longitudinal fetal growth: evidence from Mendelian randomization and parental negative control studies" (PMEDICINE-D-19-01958R1) for review by PLOS Medicine.

I have discussed the paper with my colleagues and the academic editor and it was also seen again by reviewers. I am pleased to say that provided the remaining editorial and production issues are dealt with we are planning to accept the paper for publication in the journal.

[LINK]

We look forward to receiving the revised manuscript by Oct 11 2019 11:59PM. 

Sincerely,

Clare Stone PhD

Acting Chief Editor

PLOS Medicine

plosmedicine.org

Requests from Editors:

Title – "We appreciate that mendelian randomization studies can identify potential causal pathways and that a randomized trial is not an option for this research question. However, we ask that you remove "Effects of" and “evidence” from the title to adhere to PLOS Medicine style."

As with above, and also from previous requests for revisions – please remove causal language throughout the main text. There are multiple uses of the word ‘evidence’(and ‘robust evidence at line 581), ‘causal’ and ‘effect of’ through title, abstract, Author Summary, and main text. Please remove and tone down. 

Comments from Reviewers:

[LINK]

---

## [Editor Report · Decision Letter 2]

21 Oct 2019

Dear Dr. Brand, 

On behalf of my colleagues and the academic editor, Dr. Cosetta Minelli, I am delighted to inform you that your manuscript entitled "Associations of maternal quitting, reducing and continuing smoking during pregnancy with longitudinal fetal growth: findings from Mendelian randomization and parental negative control studies" (PMEDICINE-D-19-01958R2) has been accepted for publication in PLOS Medicine. 

PRODUCTION PROCESS

PRESS

PROFILE INFORMATION

Thank you again for submitting the manuscript to PLOS Medicine. We look forward to publishing it. 

Best wishes, 

Clare Stone, PhD

Managing Editor 

PLOS Medicine

plosmedicine.org